# Compliance and Treatment Outcomes of Various Regimens for Trichomoniasis in Trinidad and Tobago

**DOI:** 10.3390/medsci7100097

**Published:** 2019-09-20

**Authors:** Aruna Kumari Divakaruni, Bisram Mahabir, F. A. Orrett, Sneha Rao Adidam, Srikanth Adidam Venkata, V. Chalapathi Rao Adidam, Srinivas Divakaruni

**Affiliations:** 1Queens Park Counseling Center and Clinic, Ministry of Health, Port of Spain, Trinidad & Tobago; arunakdiva@gmail.com (A.K.D.); bisrammahabir@hotmail.com (B.M.); 2South West Regional Health Authority, San Fernando, Trinidad & Tobago; drfao4301@yahoo.com; 3Faculty of Medical Sciences, The University of the West Indies, St. Augustine, Trinidad & Tobago; asneharao@hotmail.com (S.R.A.); sri.rao.school@gmail.com (S.A.V.); avcraouwi@gmail.com (V.C.R.A.); 4University of Massachusetts, Amherst, MA 01003, USA

**Keywords:** *Trichomonas vaginalis*, compliance, treatment, STIs, HIV, cost-effectiveness

## Abstract

Trichomoniasis is the most common non-viral sexually transmitted disease (STD) globally and yet is not a reportable disease. *Trichomonas vaginalis* is an important source of reproductive morbidity and may increase risk of acquisition and transmission of human immunodeficiency viruses (HIV). The World Health Organization (WHO) and the Control Disease Center (CDC) recommend various regimens of nitroimidazole s for treatment. The common nitroimidazoles used for trichomoniasis are metronidazole and tinidazole, which vary in their cost, efficacy, and side effect profile. It is relevant to study these factors for better management of the patients. This study aimed to compare and study the efficacy, compliance of various treatment regimens, their outcomes, and side-effects for trichomoniasis, among STI clinic attendees in Trinidad. A clinical trial study was designed, and after obtaining the informed consent, a routine clinical examination was conducted and the swabs for trichomoniasis tests were collected for diagnosis from the 692 participants. Out of 692 participants, 82 patients with positive diagnosis of *Trichomonas* infection were treated according to the patient’s choice, using different drug regimens. Compliance to treatment, side effects, and outcome were evaluated. The prevalence of trichomoniasis in the population attending our STI clinic is 11.9% and prevalence of HIV is 9%. Of the total 82 participants for the treatment, 80% were females; nearly 90% of the patients belonged to age group 15–45 years, and over 60% were below 30 years. Among those diagnosed for *Trichomonas vaginalis*, 14.6% had coexistent HIV infection. The compliance with respect to single dose treatment was significantly better than the long-duration oral regimen and has a significant relation with side effects of the treatment. The outcome is generally better and comparable and shows no significant difference between different treatment regimens used in the study. Metronidazole and tinidazole are commonly used drugs in various regimens. Compliance is better with those treated with tinidazole and metronidazole single dose than with other groups. Outcome is comparable between these regimens, especially when combined with other important factors like abstinence and treatment of the partners. The treatment regimens mainly differed in the compliance side effects profile and duration of therapy, which suggests that to improve the compliance of the drugs with fewer side effects, short course regimen would be a preferred choice.

## 1. Introduction

Trichomoniasis is a highly prevalent, treatable, non-viral Sexually Transmitted Disease (STD) of worldwide importance and is the most common curable STD. According to the World Health Organization’s (WHO) fact sheet, it is estimated that more than 143 million cases of trichomoniasis occur annually worldwide [1], and almost half of all curable sexually transmitted diseases worldwide might be attributable to *Trichomonas vaginalis* [2]. The estimated incidence in normal population is approximately 10%. Trichomoniasis has been considered one of the most common sexually transmitted diseases due to prevalence rates of 15% or higher among women in many developing countries [3]. Despite being a readily diagnosed and treated STD, trichomoniasis is not a reportable infection, and control of the infection has received relatively little emphasis from public health STD control programs. *T. vaginalis* infection is associated with two to threefold increased risk for HIV acquisition, preterm birth, and other adverse pregnancy outcomes among pregnant women [4,5,6].

Trichomoniasis is commonly treated with metronidazole and tinidazole drugs. These drugs belong to the 5-nitroimidazole drug family with 95% cure rate for *Trichomonas vaginalis*. The guidelines as per The WHO and US Centers for Control Disease (CDC) include:Metronidazole or tinidazole 2 g single dose.Metronidazole 400–500 mg Twice a day (Bis In Die, BID), 7 days dose.

Furthermore, if a patient’s single dose of metronidazole therapy fails, a 7-day dose of metronidazole or even single dose tinidazole can be administered [7].

Nitroimidazoles are inexpensive, and short treatment regimens are as effective as longer treatment regimens, which makes treatment of individual cases or even large-scale interventions quite feasible in under-resourced areas, but medication resistance for the above drugs is a worldwide concern. Further, failure to treat partners shows a lack of therapeutic success.

In one study, tinidazole was found to be more effective than metronidazole, although the study quality, comparatively, was not optimal [8].

The incidence and prevalence rates have not been clearly and reliably established in Trinidad and Tobago, especially in the high-risk population. There are no published reports about prevalence of this disease in the Caribbean Islands. Trichomoniasis causes significant morbidity, psychological stress, and economic burden on the community, and its association with transmission of HIV has cost implications in terms of time and money to the individual or the government.

### Aim of the Study

To study and describe the drug efficacy, drug compliance, treatment outcomes, and side effects of various regimens for trichomoniasis in high risk populations of Trinidad and Tobago.

## 2. Materials and Methods

### 2.1. Study Design

The study is a quasi-experimental clinical trial design, combined with a cross-sectional survey method, which is more appropriate to estimate the prevalence of an STD through laboratory conformation. Patients were recruited at the public STD disease clinic, Queens’s Park Counseling Center and Clinic (QPCC&C), Ministry of Health, Trinidad and Tobago. A convenience sample with consecutive sampling was used to recruit all males and females who were eligible, if they are over 15 years of age or had sexual exposure and consented for routine genital examination. Informed consent was obtained from all the patients recruited in the study. In the case of minors (persons below the age of eighteen), informed consent was obtained from the parent or guardian. At the time of obtaining consent, the participant was given the required information regarding the purpose of the study, treatment options, confidentiality, and rights and responsibilities. The study was approved by Medical Ethics Committee, Ministry of Health, Port of Spain, approval number 2014.

### 2.2. Sample Size

Sample size was calculated based on precision and estimated prevalence. Considering that the estimated prevalence of *Trichomonas* among high-risk populations is 20% and precision of 3% (for disease prevalence >10%), then the largest sample size needed for Trichomonas would be 683. The current study included a sample of 692 participants in total, which is slightly more than the estimated value of 683 [9].

### 2.3. Data Collection

A questionnaire that was prepared, tested, revised, and approved by Medical Ethics Committee was administered to collect the clinical data. Following a routine clinical examination, the samples were collected for laboratory testing. Vaginal/urethral swabs were collected—one for wet mount preparation of *T. vaginalis*, one for In Pouch culture, and two for OSOM rapid test, fluorescent antibody testing, and also HIV testing. When laboratory test results were positive for *T. vaginalis*, the patients were treated according the posology guidelines. After discussing medication options with the patients, they were categorized into four groups based on their most acceptable drug regimen via a quasi-experimental research method.
Group I: metronidazole 2 g single dose orally.Group II: metronidazole 400 mg twice a day for 7 days.Group III: with tinidazole 2 g single dose orally.Group IV: topical vaginal metronidazole gel/cream twice a day for 7–10 days.

The patients who received treatment were re-evaluated clinically upon follow-up with emphasis on drug compliance, side effects, abstinence, treatment of the partner(s), and efficacy by testing for organisms. Patients were asked to return 7 and 14 days after the start of therapy; and all those who returned for follow-up were tested for *T. vaginalis* organisms with repeated tests as above. Compliance to treatment, side effects, and outcome were evaluated. The patients who did not return for follow-up and test of cure were interviewed over the telephone about the clinical response and side effects.

Treatment failure was considered when reappearance of *Trichomonas* within 14 days of the start of treatment in a patient who denied sexual contact. Reinfection was defined as reappearance of *Trichomonas* in a patient who admitted further sexual contact. Recurrence was treatment failure plus reinfection. Resistance was defined as presence of continuous clinical symptoms correlated with positive tests even after treatment. Those who failed to return for follow up were contacted for assessment of response. Contact tracing was done on a few patients, and partners of *T. vaginalis*-positive patients were also treated.

Results were documented and statistical analysis was done with the help of SAS software version 9.4 (SAS Institute, Cary, NC, USA). Simple frequencies and descriptive statistics were done. The different treatment regimens used were compared using Chi-Square test and correlation studies, further confirmed by linear regression. The *p*-value was considered significant at *p* < 0.05.

## 3. Results

Out of 692 patients who were tested for *T. vaginalis* infection using z wet mount, rapid test, and in pouch culture test, 82 patients were positive for the infection. These 82 patients with established diagnosis of *Trichomonas* infection were treated using different regimens as noted above.

Among these 82 patients who were treated for Trichomonas, the majority were females. It was also observed that nearly 90% of the patients belonged to age group 15–45 years, and over 60% were below 30 years of age. Among those diagnosed positive for *T. vaginalis*, 14.6% had associated HIV infection.

Table 1 shows the prevalence rates of trichomoniasis and HIV in the population attending an STD clinic in Trinidad and Tobago. Our comparatively large study consisting of a total of 692 cases which comprised patients attending STD clinics revealed an overall prevalence of trichomoniasis of 11.9%. Gender-related trichomoniasis prevalence showed a higher rate of infection in females at 9.4%, compared to that of males, which was 2.4%. The difference in gender prevalence could be due to differences in the anatomy of genital tract and unequal number of males and females attending the clinic. The prevalence rate is much higher than the unpublished annual statistics at our clinic, which is about 3% each year for trichomoniasis.

The majority of the patients belonged to younger age groups, and a higher number of positive cases were between 15 and 39 years. In females, more numbers were recorded between 20 and 29 years, and males showed a bimodal peak at 20–24 and 35–39 years of age. The total prevalence of HIV in our sample population is 9.1%, which is much higher than the estimated prevalence of HIV in the general population [10].

Table 2 compares different groups who received treatment upon diagnosis of trichomoniasis. The various parameters evaluated are listed in the first column. Drug compliance was poor for Groups II and IV, and metallic taste was prominent with metronidazole 2 g single dose. Nausea and vomiting were prominent with a 1-week course of metronidazole. The prominent side effect for tinidazole single dose was nausea in about 13.7% of participants in the group. Metronidazole vaginal gel was used in only 6 patients, who did not report any side effects. There was no significant statistical difference for other side effects in the participant groups. In our study, the cure rates for all the groups were comparable and did not show any significant statistical difference in their efficacy. Overall results were better for drug compliance with those treated with tinidazole and metronidazole single dose than with other groups as expected.

Table 3 shows correlations between treatment, side effects, and drug compliance. Chi-square tests [11] are usually suitable to test association between categorical variables when the frequency is large. Fisher’s exact test [11] can used to test an association when the frequency is small and uses the exact numbers of the sample. For our study sample, both Chi-square and the Fisher’s exact test were done for the association between the above categorical variables. The Chi-square results show correlation between drug compliance and side effects of the treatment was significant (*p*-value < 0.05) compared to other associations as shown above, although Fisher’s exact test shows all three correlations were significant in the study sample.

The above correlations were further confirmed with linear regression, where drug compliance was considered as the dependent variable on side effects and treatment modality. The model showed relation between side effects and drug compliance in the study as statistically significant.

Table 4 below shows the drug efficacy as (absolute risk = negative events/total events or trichomoniasis negativity/total patients) negative test percentage and frequency in treated patients. The negative tests are based on combined results of the trichomoniasis rapid test, wet mount test, culture (in pouch), and direct fluorescent antibody test. Even if one test was positive, the combined results were considered as positive, and when all the four tests were negative, the tests were considered as negative.

## 4. Discussion

Nitroimidazoles is the mainstay in the treatment of trichomoniasis. Metronidazole is the most common drug used for trichomoniasis since it is less expensive than tinidazole [12]. Metronidazole is relatively cheap and is widely available in public health institutions in Trinidad, and tinidazole is comparatively more expensive and less widely available. Presently, different options are available for the treatment of trichomoniasis. It may be noted that over the years, there has been gradual reduction in the duration of treatment with a single dose of metronidazole or tinidazole giving satisfactory results.

The current study compared various treatment regimens used in trichomoniasis. It compared the efficacy of various strategies for the treatment of trichomoniasis particularly, short versus long oral treatment regimens, compliance, and side effects. The overall efficacy of treatment options for oral nitroimidazole was satisfactory and compared well with other studies. The cure rates ranged from 90% to 100%, and similar cure rates were observed by Csonka (1971), Woodcock (1972), and Morton, Sawyer et al. (1976) [13,14,15,16,17,18]. There was no significant difference between the oral nitroimidazole treatment options [19]. Although fewer numbers of cases were studied, the efficacy of vaginal application of metronidazole was comparatively high. The results of a few previous studies reported that vaginal application and its effectiveness is unsatisfactory and undefined [20].

We observed that compliance with respect to single dose treatment was significantly better than long-duration oral regimen and vaginal application. An advantage of this would be that the patient receives the drug under supervision or has less risk of missing the dose that may occur with a long-duration regimen. Single dose administration is an advantage in the treatment of outpatients and also convenient for patients and their partners [21,22].

Side effects of oral nitroimidazoles are previously well documented. They include a bitter metallic taste, nausea, vomiting, a disulfiram-like reaction with ingestion of alcohol, and dizziness. Despite the low doses required for treatment of trichomoniasis, side effects occurred in more than half of the patients, especially with the metronidazole 2 g dose. This research study reports that 40%–55% of patients receiving oral nitroimidazole had side effects. The common side effects were of gastrointestinal origin and comprised metallic taste, nausea, vomiting, and anorexia. Metallic taste was more significant with single oral dose of metronidazole than long-duration oral metronidazole or tinidazole. Overall, other side effects were more prominent with metronidazole than tinidazole and compared well with reports and reviews. Some of the previous studies on single dose treatment showed that tinidazole is curative at lower doses than metronidazole. The therapeutic doses of tinidazole resulted in fewer and milder side effects [23,24,25]. In our study, the final tests showed a comparative response with metronidazole and tinidazole. The main difference between different treatment regimens was observed in drug compliance and side effect profile.

It is estimated that perhaps as few as 20% of partners receive treatment [23]. The majority of the *Trichomonas*-positive patient partners were treated in our study through contact tracing protocol. Failure to treat partners may lead to apparent lack of therapeutic success and, because trichomoniasis is an STD, treatment of sex partners must be part of the treatment regimen of infected participants. Contact tracing should be undertaken, and all resulting sexual contacts attending should be treated for *T. vaginalis* regardless of the results of their investigations.

### 4.1. Treatment Failure

Only two cases were found to be nonresponsive to treatment, and other cases were considered as either as reinfection or recurrence. The two cases were confirmed to be drug-compliant and followed abstinence for the required period. These patients were considered as resistant cases. This problem was seen with the 7-day treatment by metronidazole. Treatment failure in some patients who were prescribed the 7-day course may have been due to failure to take the treatment as directed. Some cases classed as re-infections may have been due to treatment failure [22,26]. Metronidazole resistance is an emerging problem, but its clinical importance is not yet clear [22,27].

There are conflicting data on the use of metronidazole during pregnancy. Some studies claim possible teratogenicity and other adverse effects, like prematurity. However, the association has been denied by others [3,28]. Vaginal application of metronidazole can be used in those who were pregnant or possibly pregnant and in those who preferred the method. According to a recent study conducted by Thu et al., treatment of trichomoniasis with metronidazole in pregnant patients may break the endosymbiotic relationship with *Mycoplasma hominis*, which can cause further morbidity [29]. The authors suggested molecular techniques for diagnosis in such cases.

### 4.2. Cost Effectiveness of Various Regimens

Nitroimidazole is inexpensive (the mean cost of generic tinidazole is US$0.04/500 mg tablet, or US$0.16 for a typical 2 g dose) [2]. This feature, combined with the fact that short treatment regimens (typically single dose) are highly effective, makes treatment of individual cases or even large-scale interventions quite feasible in under-resourced areas [3,26,30]. Tinidazole single dose is more cost-effective compared to other treatment regimens for the sponsor, but outcome is not greater than other regimens. Cost effectiveness is not an important factor for the patients in this trial because the medicines were sponsored by the government.

### 4.3. Strengths and Weaknesses

A convenient sampling method was used in recruiting the *T. vaginalis* patients because it is a practical and cheaper way of recruiting study participants and uses an existing infrastructure, such as clinic facilities and staff. This is further supported by the fact that this study established *T. vaginalis* prevalence in high risk population coming to an STD clinic and did not attempt to establish *T. vaginalis* prevalence in the general population. The power of the study calculated is good for high prevalence in the STD clinic participants. The study chose a quasi-experimental method in placing the participants according to their choice in treatment groups because we considered that a strict experimental method is unethical, especially when there was previous evidence showing the difference in side effects of treatment options, and we followed patients’ choice. This resulted in unequal number of participants in treatment groups but did not affect the quality of the study as shown in the statistics.

## 5. Conclusions

Trichomoniasis is the most common nonviral STD globally, and it is not a reportable disease. *Trichomonas vaginalis* is an important source of reproductive morbidity and may increase risk of acquisition and transmission of HIV. Nitroimidazoles are the only class of antimicrobial medications known to be effective against *T. vaginalis* infections. Of these drugs, metronidazole and tinidazole are commonly used in various regimens. In this study, overall results were comparable in those treated with tinidazole and in other groups. Side effects of medication were more prominent with metronidazole than tinidazole. tinidazole is more expensive than metronidazole, and it is not widely available. Only two cases were found to be nonresponsive to treatment. The association between drug compliance and side effects of the various medication were confirmed in our study. To make the patient more compliant with treatment, it is always better to use the drug with fewer side effects and of a shorter regimen.

## Figures and Tables

**Table 1 medsci-07-00097-t001:** Comparison of demographic variables like sex, age and HIV positivity in the participants of the study.

Variable	Sub-Category	Trichomoniasis Positive (*n* = 82)		Trichomoniasis Negative (*n* = 610)	
		Frequency	% in Total Sample	Frequency	% in Total Sample
Sex	Male	17	2.4	249	35.6
	Female	65	9.4	361	52.1
Age (in years)	5–14	0	0	9	1.3
	15–24	33	4.8	243	35.1
	25–34	27	3.9	197	28.5
	35–44	18	3.2	83	12
	45–54	3	3.6	59	8.5
	55–64	1	0.15	15	2.2
	>65	0	0	4	0.6
HIV	Positive	12	1.7	51	7.4
	Negative	70	10.1	559	80.8

HIV: Human Immunodeficiency Virus.

**Table 2 medsci-07-00097-t002:** Comparison of different treatment options for trichomoniasis which were treated.

Variable	Group I	Group II	Group III	Group IV
No. of cases (*n* = 82)	18	36	22	6
Drug compliance	100%	61.1%	100%	33.3%
Abstinence	33.3%	36.1%	27.27%	16.67%
Cure rate	94.4%	97.2%	90.9%	100%
Side effects	55.6%	50%	31.8%	0%
Nausea	38.9%	27.8%	13.7%	0%
Vomiting	22.2%	30.6%	4.6%	0%
Anorexia	16.7%	13.9%	9.1%	0%
Metallic taste	44.4%	25%	2%	0%
Abdominal cramps	5.6%	13.9%	0%	0%
Loose stools	0%	3%	0%	0%
Headaches	5.6%	13.9%	0%	0%
Rash	0%	5.6%	0%	0%

Group I: metronidazole 2 g single dose orally. Group II: metronidazole 400 mg twice a day for 7 days. Group III: with tinidazole 2 g single dose orally. Group IV: topical vaginal metronidazole gel/cream twice a day for 7–10 days.

**Table 3 medsci-07-00097-t003:** Correlation between treatment, compliance and side effects.

Statistics	Correlations (Probability, *p*-Value)		
	Treatment vs. Side Effects	Treatment vs. Compliance	Compliance vs. Side Effects
Chi-square	0.0566	0.0867	<0.0001
Fishers exact test	0.0004	<0.0001	<0.0001

**Table 4 medsci-07-00097-t004:** Comparison of efficacy of drug regimens.

Absolute Risk	Group I	Group II	Group III	Group IV
Drug efficacy percentage (AR)	94.4	97.2	90.9	100
Confidence intervals for AR	92.1–96.7	95.6–98.8	88.8–93	96–104
Drug efficacy frequency	17/18	35/36	20/22	6/6

AR: absolute risk.

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
