# Peer review of "Compliance and Treatment Outcomes of Various Regimens for Trichomoniasis in Trinidad and Tobago"

_medsci, 2019, doi:10.3390/medsci7100097_

Round 1

Reviewer 1 Report

The manuscript reflects the results obtained after the development of a comparative study in which interesting determinations concerning the efficacy of different posology, the treatment during pregnancy, the compliance to treatment and side effects. Also the incidence detected in this populations and the high correlation with HIV coinfection is another emergency alarm that should be taken into consideration by the health community organisms and governments.

Although the study is interesting there are many format errors that must be changed:

- Trichomonas vaginalis: the name of the specie "vaginalis" MUST be in lower case. For example: line 14, keywords, 98, 225, etc

- Trichomonas vaginalis, T. vaginalis or Trichomonas MUST be in italics. Lines: 14, 24, 45, 89, 90 etc

-Nitro-Imidazole should be changed for nitroimidazole. Ex: lines 16, 54, 60, 168, 157, 176, 180, 206

- Trichomoniasis is the denomination of a disease and should be written in lower case "trichomoniasis". Lines 17, 20, 22, 26, 44, 74, etc

- Please change in line 56 TNZ 2 gm for "TNZ 2 g single dose"

-If the authors decide to use abbreviations for Trichomonas vaginalis (TV), metronidazole (MDZ) or tinidazole (TDZ) please, use those abbreviations throughout the manuscript. And the name of both drugs should be also in lower case (metronidazole and tinidazole). Lines: 53, 64, 137, 138, etc

- Please explain the concept "stat" table 2, lines 34, 101, 103, 135.

- Line 23: use the number of patients in numbers or in letters but it is not relevant to write Eighty two (82).

Regarding other aspects not related with format of specific grammar changes are suggested below:

ABSTRACT: This reviewer considers that the sentence from line 16 to 18 could be re-write just in order to give more importance to the study. Perhaps in two sentences instead of only one would call the reader´s attention.

INTRODUCTION:

Some reference are missed (line 65). 

The last sentence of the introduction (lines 70 to 72) is the same as the aim of the study so it can be removed or changed.

MATERIALS AND METHODS:

Line 77: Comas are missed "The study is a quasi-experimental clinical trial design, combined with cross sectional survey method, which is more appropriate..."

Line 89: "Considering THAT the estimated prevalence of Trichomonas among high risk population is 20% and precision of..."

Line 97: The word treat or treatment is used at least 5 times in the next paragraph, please avoid repeating the same word and use synonymous (medication, posology, attend, etc)  "When laboratory test results were positive for Trichomonas vaginalis, the patients were treated according the treatment guidelines. After discussing treatment options with the patients, they were categorized into four (4) groups based on their most acceptable treatment regimen by quasi-random method."

Line 105: The next comment should be in the discussion section and not in the M&M "Metronidazole is relatively cheap and is widely available in public health institutions in Trinidad. Tinidazole is comparatively more expensive and is less widely available."

RESULTS:

The most relevant results observed in the tables could be more described in detail in the text, including the exact percentages when comparing groups.

This reviewer considers that the tables related with statistical results (tables 3, 4a and 4b) could be removed and include a brief explanation of the statistical results obtained.

DISCUSSION:

This reviewer misses the reference that supports line 157 "Metronidazole is the most commonly used drug worldwide"

Also the references that underwrite the cura rates of 5-nitroimidazoles are from 40-50 years ago, new references should be included, and also resistant cases must be mentioned.

The authors explain in detail the possible side effects when using 5-nitroimidazoles for treatment, however no relation with the results obtain in the research are compared or mentioned. Also this reviewer will recommend to include the reference that indicates that " Despite the low doses required for treatment of trichomoniasis, side effects occur in more than half of patients, " line 178.

In relation with pregnancy, discussed from line 202 to 205, I strongly recommend to revise and include in the discussion section the article of Trung Thu et al 2018, Trichomonas vaginalis transports Mycoplasma hominis and transmits the infection to human cells after metronidazole treatment: a potential role in bacterial invasion of fetal membranes and amniotic fluid. Journal of Pregnancy ID Article 5037181 

REFERENCES:

This reviewer is missing more modern reviews.

Author Response

The manuscript reflects the results obtained after the development of a comparative study in which interesting determinations concerning the efficacy of different posology, the treatment during pregnancy, the compliance to treatment and side effects. Also the incidence detected in this populations and the high correlation with HIV coinfection is another emergency alarm that should be taken into consideration by the health community organisms and governments.

Reply: Thanks for your comments.

Although the study is interesting there are many format errors that must be changed:

- Trichomonas vaginalis: the name of the specie "vaginalis" MUST be in lower case. For example: line 14, keywords, 98, 225, etc

Reply: Agreed, changed manuscript according to suggestions.

- Trichomonas vaginalis, T. vaginalis or Trichomonas MUST be in italics. Lines: 14, 24, 45, 89, 90 etc

Reply: Agreed, changed manuscript according to suggestions.

-Nitro-Imidazole should be changed for nitroimidazole. Ex: lines 16, 54, 60, 168, 157, 176, 180, 206

Reply: Agreed, changed manuscript according to suggestions.

- Trichomoniasis is the denomination of a disease and should be written in lower case "trichomoniasis". Lines 17, 20, 22, 26, 44, 74, etc

Reply: Agreed, changed manuscript according to suggestions.

- Please change in line 56 TNZ 2 gm for "TNZ 2 g single dose"

Reply: Agreed, changed manuscript according to suggestions.

-If the authors decide to use abbreviations for Trichomonas vaginalis (TV), metronidazole (MDZ) or tinidazole (TDZ) please, use those abbreviations throughout the manuscript. And the name of both drugs should be also in lower case (metronidazole and tinidazole). Lines: 53, 64, 137, 138, etc

Reply: Agreed. Short drug names were replaced with long names in lower letters throughout the manuscript.

- Please explain the concept "stat" table 2, lines 34, 101, 103, 135.

Reply: Agreed. Explained stat concept.

- Line 23: use the number of patients in numbers or in letters but it is not relevant to write Eighty two (82).

Reply: Agreed. Changed manuscript according to suggestions.

Regarding other aspects not related with format of specific grammar changes are suggested below:

ABSTRACT: This reviewer considers that the sentence from line 16 to 18 could be re-write just in order to give more importance to the study. Perhaps in two sentences instead of only one would call the reader´s attention.

Reply: Agreed, changed manuscript according to suggestions.

INTRODUCTION:

Some reference are missed (line 65).

Reply: Agreed, added reference to line 65.

The last sentence of the introduction (lines 70 to 72) is the same as the aim of the study so it can be removed or changed.

Reply: Agreed, removed the repeating sentence from introduction.

MATERIALS AND METHODS:

Line 77: Comas are missed "The study is a quasi-experimental clinical trial design, combined with cross sectional survey method, which is more appropriate..."

Reply: Agreed, changed manuscript according to suggestions.

Line 89: "Considering THAT the estimated prevalence of Trichomonas among high risk population is 20% and precision of..."

Reply: Agreed, added ‘that’ to the sentence at line 89.

Line 97: The word treat or treatment is used at least 5 times in the next paragraph, please avoid repeating the same word and use synonymous (medication, posology, attend, etc)  "When laboratory test results were positive for Trichomonas vaginalis, the patients were treated according the treatment guidelines. After discussing treatment options with the patients, they were categorized into four (4) groups based on their most acceptable treatment regimen by quasi-random method."

Reply: Agreed, changed the line 97 according to suggestions.

Line 105: The next comment should be in the discussion section and not in the M&M "Metronidazole is relatively cheap and is widely available in public health institutions in Trinidad. Tinidazole is comparatively more expensive and is less widely available."

Reply: Agreed, changed manuscript according to suggestions.

RESULTS:

The most relevant results observed in the tables could be more described in detail in the text, including the exact percentages when comparing groups.

Reply: Agreed, more description of the tables were added.

This reviewer considers that the tables related with statistical results (tables 3, 4a and 4b) could be removed and include a brief explanation of the statistical results obtained.

Reply: Tables 4a and 4b removed and description was added. Table 3 findings were described in detail in the next paragraph.

DISCUSSION:

This reviewer misses the reference that supports line 157 "Metronidazole is the most commonly used drug worldwide"

Reply: Agreed, added reference to line 157.

Also, the references that underwrite the cure rates of 5-nitroimidazoles are from 40-50 years ago, new references should be included, and also resistant cases must be mentioned.

Reply: Agreed, added new references for the cure rates and resistant cases were mentioned.

The authors explain in detail the possible side effects when using 5-nitroimidazoles for treatment, however no relation with the results obtain in the research are compared or mentioned.

Reply: Agreed, changed manuscript according to suggestions.

 Also this reviewer will recommend to include the reference that indicates that " Despite the low doses required for treatment of trichomoniasis, side effects occur in more than half of patients, " line 178.

Reply: Agreed, the line is changed so that it reflects our study findings rather than metronidazole in general.

In relation with pregnancy, discussed from line 202 to 205, I strongly recommend to revise and include in the discussion section the article of Trung Thu et al 2018, Trichomonas vaginalis transports Mycoplasma hominis and transmits the infection to human cells after metronidazole treatment: a potential role in bacterial invasion of fetal membranes and amniotic fluid. Journal of Pregnancy ID Article 5037181

Reply: Agreed. Suggested reference was added in relation with pregnant patients and trichomoniasis.

REFERENCES:

This reviewer is missing more modern reviews.

Reply: Agreed. We have added some modern references to the manuscript.

Thank you for reviewing our manuscript. Your feedback and suggestions were very well received among our authors and we really appreciate them. Thank you for your support.

Reviewer 2 Report

This is a comparison of 2 g dose of metronidazole versus 500 mg BID dose for the treatment of trichomoniasis.  There are several issues that need to be addressed:

The authors do not present the findings in the standard RCT format.  No CONSORT table is presented.  No comparison of baseline characteristics by arm (to examine if randomization worked).  Was the trial registered on clinicaltrial.gov or one of the clinical trial registries? 

There are many typo’s in the paper which are very distracting.  Trichomonas vaginalis is misspelled throughout and not italicized as it should be.  The results should be put in the past tense as the study is completed.

The randomization method is not well described.  What do the authors mean by quasi-experimental?  Were the subjects or persons who examined the outcome blinded?

The inputs for the sample size are not clear.  It should be based on what they think the outcome will be in the control arm and the difference they expect to see.   The authors did not appear to account for four arms in their calculation.  It is doubtful that the authors had sufficient power to detect differences in four arm with n=82.

Table 1 is not helpful.  They should describe the group at baseline by arm.    In Table 5, the confidence intervals should be presented.   I do not understand the significance of tables 1-4b.  Did the authors do an intent-to-treat analysis?

A poor study design is also an ethical concern.

Author Response

Comments and Suggestions for Authors

This is a comparison of 2 g dose of metronidazole versus 500 mg BID dose for the treatment of trichomoniasis. 

Reply: Please note there are 4 treatment groups.

There are several issues that need to be addressed:

The authors do not present the findings in the standard RCT format.  No CONSORT table is presented.  No comparison of baseline characteristics by arm (to examine if randomization worked).  Was the trial registered on clinicaltrial.gov or one of the clinical trial registries? 

Reply: Please note this is not a trial based on RCT. This study is based on quasi-experimental design in which there is no randomization. Patients were given the choice to choose any one of the treatment choices after explaining the treatment, side effects, their access to meds and the availability of meds. The controlling factor in this study is not randomization, but it is based on understanding and availability of the meds to the participants and their choice. The study is not about new drugs in the market, but finding the differences between established meds as the treatment choices. The baseline characters checked were positivity to Trichomoniasis. Several other factors at baseline were also checked in regards to their test positivity, but as this paper is mainly focused on treatment modalities of Trichomoniasis, the results are mainly focused on treatment choices. The trial is registered and approved by Ministry of health Trinidad and Tobago Ethical committee. The approval number and consent was already provided.

There are many typo’s in the paper which are very distracting.  Trichomonas vaginalis is misspelled throughout and not italicized as it should be.  The results should be put in the past tense as the study is completed.

Reply: Agreed and changes were done accordingly.

The randomization method is not well described.  What do the authors mean by quasi-experimental?  Were the subjects or persons who examined the outcome blinded?

Reply: As discussed above, this is not an RCT. This is a quasi-experimental design study in which there is no blinding either from the investigators or participants. Instead of randomization, the choice of patients is used as the controlling factor. A quasi experimental design study is otherwise called non-randomized trial, in which there is no randomization[1]. We believed this type of study is suitable for the community because it is more ethical, especially when using already established medications.

The inputs for the sample size are not clear.  It should be based on what they think the outcome will be in the control arm and the difference they expect to see.   The authors did not appear to account for four arms in their calculation.  It is doubtful that the authors had sufficient power to detect differences in four arm with n=82.

Reply: As discussed above the study is not an RCT.  Out of about 692 patients, 82 patients who are positive for Trichomonas were treated according to their treatment of choice. The statistical significance for 82 patients and treatment groups were discussed in the results.The significance of power might be irrelevant in a completed study, where we were not trying to prove any cause and effect theories, but only the association or correlation between drug efficacy, compliance and side effects of the drug regimens.

Table 1 is not helpful.

Reply: The table 1 is an introduction to the demography of the STI population in the study, which shows prevalence values of trichomoniasis and HIV in the STI population.  

They should describe the group at baseline by arm.

Reply: All the treated participants have positive tests for trichomoniasis at baseline. Drug compliance, side effects and treatment outcome can be studied only after the treatment.   

 In Table 5, the confidence intervals should be presented.

Reply: Agreed, added confidence intervals.

I do not understand the significance of tables 1-4b.

 Reply: The tables are changed and added more explanation.

Did the authors do an intent-to-treat analysis?

 Reply: As discussed above this study is not an RCT. If the test was positive, we treated the participants according to the participant’s choice.

A poor study design is also an ethical concern.

Reply: Agreed, but the authors think this comment will not apply to our study.

The reviewer compares this to an RCT and made comments to that effect. But this is not RCT. In fact, an RCT in case of established drugs in the market, will be unethical as the choice of treatment is not given to the participants. Investigators may differ in their opinion but a quasi-experimental study based on participant’s choice is much more ethical than any RCT, since there is no choice for participants in RCT.  This study is perfectly ethical as all the criteria for an ethical study like taking consent, ethical committee approval were fulfilled. The participants were given the choice of their treatment, which perfectly was in accordance with deontological perspective of clinical research. We believe, doing an RCT study with the same choices of meds for the same disease in contemporary conditions might be unethical as the choice is not given to the participants.

Thank you for reviewing our manuscript. Your feedback and suggestions were very well received among our authors. We really appreciate your time and effort in this review. Thank you for your support.

[1] Harris AD, McGregor JC, Perencevich EN, et al. The use and interpretation of quasi-experimental studies in medical informatics. J Am Med Inform Assoc. 2006;13(1):16–23. doi:10.1197/jamia.M1749

Reviewer 3 Report

As the authors themselves state, the presented data are especially relevant for the studied population, for which there exist no previous ones. As such, the results may have clinical implications for women in that area, is at the same time interesting epidemiologically in a general sense, and therefore merits publication. Of course, the design is not perfect, since a randomized prospective study would have been the ideal one, but given the circumstances, it is more than acceptable and might be useful in practice for the addressed population. Some minor grammatical corrections are necessary

Author Response

Reviewer: As the authors themselves state, the presented data are especially relevant for the studied population, for which there exist no previous ones. As such, the results may have clinical implications for women in that area, is at the same time interesting epidemiologically in a general sense, and therefore merits publication. Of course, the design is not perfect, since a randomized prospective study would have been the ideal one, but given the circumstances, it is more than acceptable and might be useful in practice for the addressed population. Some minor grammatical corrections are necessary.

Reply: Thanks for your review of our manuscript. We really appreciate your views and comments of the study. Your comments are very well received among our authors and we agree about the importance of the study to the population in which the study was done. The manuscript is changed according to your suggestions about grammatical errors.